# Development of a Parturition Detection System for Korean Native Black Goats

**DOI:** 10.3390/ani14040634

**Published:** 2024-02-16

**Authors:** Heungsu Kim, Hyunse Kim, Woo H. Kim, Wongi Min, Geonwoo Kim, Honghee Chang

**Affiliations:** 1Division of Animal Science, Gyeongsang National University, Gyeongsangnam-do, Jinju 52828, Republic of Korea; gde012@naver.com (H.K.); h3h1358@gmail.com (H.K.); 2College of Veterinary Medicine, Gyeongsang National University, Gyeongsangnam-do, Jinju 52828, Republic of Korea; woohyun.kim@gnu.ac.kr (W.H.K.); wongimin@gnu.ac.kr (W.M.); 3Department of Biosystem Engineering, Gyeongsang National University, Gyeongsangnam-do, Jinju 52828, Republic of Korea; 4Institute of Agriculture and Life Science, Gyeongsang National University, Gyeongsangnam-do, Jinju 52828, Republic of Korea

**Keywords:** goat kid, management, accelerometer, lying behavior, labor pain

## Abstract

**Simple Summary:**

Many studies on parturition detection have been conducted on large ruminants, but only a few have been conducted on small ruminants. The parturition detection system for Korean native black goats developed in this study may detect labor pains in goats and assist in the timely initiation of kid nursing to reduce stunted growth and mortality. However, applying this system to other Korean Native Black Goats or other breeds of goats will require further research due to different breed characteristics and specification conditions.

**Abstract:**

Korean Native Black Goats deliver mainly during the cold season. However, in winter, there is a high risk of stunted growth and mortality for their newborns. Therefore, we conducted this study to develop a KNBG parturition detection system that detects and provides managers with early notification of the signs of parturition. The KNBG parturition detection system consists of triaxial accelerometers, gateways, a server, and parturition detection alarm terminals. Then, two different data, the labor and non-labor data, were acquired and a Decision Tree algorithm was used to classify them. After classifying the labor and non-labor states, the sum of the labor status data was multiplied by the activity count value to enhance the classification accuracy. Finally, the Labor Pain Index (LPI) was derived. Based on the LPI, the optimal processing time window was determined to be 10 min, and the threshold value for labor classification was determined to be 14 240.92. The parturition detection rate was 82.4%, with 14 out of 17 parturitions successfully detected, and the average parturition detection time was 90.6 min before the actual parturition time of the first kid. The KNBG parturition detection system is expected to reduce the risk of stunted growth and mortality due to hypothermia in KNBG kids by detecting parturition 90.6 min before the parturition of the first kid, with a success rate of 82.4%, enabling parturition nursing.

## 1. Introduction

Korean native black goats (KNBGs) are a seasonal breed that bears its young during the colder months of fall, winter, and spring [1,2]. In the absence of postpartum care, winter parturition can result in stunted growth and mortality in newborns [3]. Studies of goats from other breeds have shown that mortality in newborn goats can be high, ranging from 7.8% [4] to 37.0% [5] during the first 1–3 days of life, and the younger the goat, the higher the mortality [6]. In sheep, early parturition nursing, colostrum feeding, and lamb drying increase survival in winter [7]. Thus, parturition nursing increases kid survival and allows for quick action even when the mother goat is not engaging in maternal behavior.

Common goat behaviors include standing, feeding, walking, watering, resting, mating, agonistic behavior, and licking. However, pre-parturient goats exhibit decreased feeding behavior, slower movement, frequent urination [8], isolation from other goats, increased lying, and changes in posture [9]. Of these, the most important are posture changes and increased lying behavior. Lying refers to the behavior of a mother goat lying on her side with her legs stretched out and constantly moving when she feels labor pains just before parturition. 

Acceleration sensors can be used to detect changes in behavior. Acceleration sensors observe values due to the acceleration of gravity and inertial acceleration due to movement. They can observe movement as data values by observing changes in multiple axes depending on the number of axes of the sensor [10,11,12]. Sheep are sometimes fitted with triaxial accelerometer sensors on their ears to collect data. Random Forest machine learning (ML) has been employed to detect lying behavior, a primary pre-parturition behavior, with up to 83.6% accuracy [13]. Moreover, Support Vector Machine (SVM) algorithms have been utilized to classify grazing, lying, standing, and walking behaviors with 76.9% accuracy [14]. In addition, activity and inactivity can be classified with 98.1% accuracy using a Decision Tree (DT) [14], and horse colic can be detected by attaching a triaxial acceleration sensor to their withers to collect data and measure rolling and sternal recumbency [15]. These studies have shown that it is possible to use various ML techniques to detect labor behaviors by attaching a triaxial acceleration sensor to the hind leg of a mother goat to measure their activity and the amount of time spent lying on their side and stretching their legs when feeling labor pains and to classify labor and non-labor behaviors. 

This study assumed the following: (1) KNBGs lie on their side, stretch their legs, and move when they feel labor pains just before parturition; (2) a triaxial accelerometer can classify and measure their behaviors, including lying on their side, stretching their legs, and moving when feeling labor pains just before parturition; (3) a Labor Pain Index (LPI), which combines the amount of side-lying and leg-extending behavior with the amount of activity, can classify labor and non-labor behavior; and (4) labor behavior can be detected using an LPI.

Therefore, the objectives of this study were to:Classify labor and non-labor behaviors using an ML classification algorithm for KNBGs;Develop an LPI that combines the amount of labor behaviors and the amount of activity to indicate the intensity of labor pain;Use the LPI to detect parturition; andDevelop a notification system that can provide early warning for labor.

## 2. Material and Methods

### 2.1. Animals and Housing 

The experiments were conducted at a traditional black goat house (35°20′62.8″ N, 128°13′70.2″ E) on the livestock farm of Gyeongsang National University, Jinju, Gyeongsangnam-do, Republic of Korea. The animals were 17 pregnant female KNBGs. 

The KNBGs’ house was an open-sided barn with a room size of 2.3 m × 7.7 m, with passageways both inside (2.3 m × 3.0 m) and outside (2.3 m × 4.7 m). 

The KNBGs remained in the same rooms in which they were previously housed to minimize stress due to rank order within the herd [16] and were managed customarily.

Feed was a combination of concentrates and hay fed at 09:10 and 14:30, and drinking water was provided in a cylindrical water trough on the outside of the barn with free access for drinking.

Herd size was determined by placing a maximum of 4 KNBGs per room to reduce rank battles due to overlarge herd sizes [17].

The stocking density was 4.4275 m^2^/hd, with at least 3 m^2^ of space allocated for pregnant goats to reduce stress [18].

### 2.2. Behavioral Observations

The behavior of the animals was recorded by installing a total of eight cameras (DS-2CD2066G2-I, HIKVISION, China) and two recorders (DS-7204HQHI-K1, HIKVISION, China) inside and outside the KNBG house.

The RGB cameras used for behavioral observation had a horizontal field of view (FOV) of 105.1°, a vertical FOV of 53.4°, a diagonal FOV of 128.6°, and a resolution of 6 MP.

All recorded data were backed up once per week during the observation period to prevent data loss due to the poor environment inside the barn (i.e., the presence of dust, humidity, and gas).

### 2.3. Attaching the Triaxial Acceleration Sensor

The triaxial acceleration values were collected using a sensor (iBS03, INGICS TECHNOLOGY, Taiwan). The sensor size was 43 mm × 43 mm × 14.8 mm, and the data communication rate was 1 Hz. The triaxial acceleration sensor was covered with a plastic housing with dimensions of 85 mm × 75 mm × 25 mm to prevent breakage and for easy and secure attachment. The triaxial acceleration sensor weight was 25 g and plastic housing was 100 g.

A triaxial accelerometer was used to analyze various behaviors of the KNBGs kept in the barn. Acceleration sensors were attached to the hind legs of the goats based on previous studies that measured the lying behavior of goats [19,20].

The hock is the joint between the tarsal bone and the tibia in a goat’s hind limb. Anatomically, the tarsal bone below the hock is cylindrical in shape, which makes it easier for sensors to slip out of place when attached. However, the tibia, located just above the hock, has muscles called the tibialis caudal and gastrocnemius muscles [21], making this area relatively flat and enabling a sensor to be affixed easily. Thus, the acceleration sensor was attached to the tibia using double-sided Velcro. The tibia has also been used to study goat gait characteristics [22] and to assess gait [23] in goats. Based on these previous studies, attaching the sensor to the tibia would allow us to assess movement accurately, as shown in Figure 1.

However, the double-sided Velcro used to attach the sensor could rub against the flesh of the tibia over time, causing crushing or wounding. Therefore, the area where the double-sided Velcro and the flesh of the tibia were in direct contact was covered with a soft cloth to prevent any friction between the Velcro and the flesh from causing wounds. 

The X-axis of the triaxial acceleration sensor was oriented up and down the goat’s body, the Y-axis was oriented left and right, and the Z-axis was oriented in front and behind the goat, as shown in Figure 2. Therefore, it was expected that the Z-axis would change significantly and that continuous movement would occur in instances of the goats’ labor behavior of lying (lateral recumbency) with their legs stretched out while feeling labor pains compared to the general behavior of standing or sitting. Thus, it was determined that the labor behavior could be measured based on the Z-axis acceleration and movement values.

The triaxial acceleration sensors were attached to the KNBGs 2–3 weeks before parturition was expected so that the KNBGs could become accustomed and adapt to them so that the sensation of a suddenly attached sensor would not distort their parturition behavior. 

### 2.4. Acceleration Data Collection

The KNBG parturition detection system consisted of triaxial acceleration sensors, gateways, a cloud server, and alarm terminals.

The triaxial acceleration sensor data were transmitted to the gateway (IGSIS, INGICS TECHNOLOGY, Taiwan)(https://www.ingics.com/ (accessed on 12 February 2024)) through Bluetooth communication, and the gateway transmitted the data to the server through Internet communication.

The cloud server (Wuyang cloud server, Wuyang corporation, Republic of Korea) stored the following acceleration data: data number, gateway number, time, received signal strength indicator (RSSI), battery life, activity or inactivity, and X-, Y-, and Z-axis acceleration values.

The alarm terminal can be used by any device that can use the Internet, and it is configured to access the cloud server to obtain each value and view it via the Internet or through an application. 

Due to the characteristics of the triaxial acceleration sensor, acceleration data were collected once a second when there was movement (activity) and once every 10 s when there was no movement (inactivity). Therefore, depending on the acceleration data during a particular time period, it would be possible to determine whether there was activity or inactivity during that time.

The acceleration data related to labor behavior was set to indicate when the animal performed additional behaviors, such as lying (1 recumbency) and stretching its legs during the behavioral observation or howling while lying (lateral recumbency). A dataset was created with the sensor values for those particular observation times. In contrast, the data related to non-labor behavior were set to indicate when the animal performed general behavior (walking, sitting, feeding, and inactivity) during the behavioral observation time period, and a dataset was created with the sensor values for those particular observation times. 

### 2.5. ML for Labor Pain Classification

The labor pain dataset was verified by observing the recordings of the 17 KNBGs engaged in labor behavior. The sensor values at the time of labor behavior obtained through behavioral observation were extracted from the server to create a dataset based on 3938 labor pain data points. For the non-labor dataset, 3938 non-labor behavioral datasets containing behaviors other than labor behavior were created for the same 17 KNBGs from the data 12 h before the parturition time of the first kid. We labeled labor pain data with a 1 and non-labor data with a 0. These labeled data were then used as the final dataset to classify labor pain and non-labor behaviors in KNBGs. 

The ML performance was compared using an SVM, a linear model, and a DT, a non-linear model. We used 80% of the total dataset as training data and the remaining 20% as test data to train with both models. We also used data augmentation to mix the data to prevent overfitting of the model to ensure accuracy.

Considering the results of the scatter plots for the main variables, the model with the most minor overlap between the linear and non-linear labor behavior classification models was determined to be the best classification model.

Variable selection is an essential process in ML and is crucial for optimizing a model’s performance and removing unnecessary information. In complex datasets, every variable may contain valuable or unnecessary data, and these unnecessary variables can lead to poor model performance and difficulty in interpretation. 

We used Weka software (Version 3, The University of Waikato, New Zealand) for variable selection. The levels of importance of the X-, Y-, and Z-axis acceleration values were evaluated using information gain, which assesses how well the data are separated by the corresponding attributes contained in the dataset, and gain ratio, which normalizes the information gain to account for the variability of the attributes.

### 2.6. Calculation of the LPI

Labor pain detection in seconds uses accelerometer data, which are prone to error because the system detects changes in posture over a short period of time. However, when the KNBGs felt pain during labor, they stayed in the lying (lateral recumbency) position longer. Therefore, the time windows collected were 8, 10, 12, and 14 min, and LPIs were calculated for each of these periods. From these calculations, we constructed a labor pain dataset, which combined the labor pain dataset from the 17 KNBGs during labor behavior with the non-labor dataset of those same KNBGs during non-labor. 

The standard range for acceleration data is between 0 and 255, but the data in this dataset occasionally contained values below 0 or above 256, which were determined to be outliers and were removed. Additionally, to improve the accuracy of the data distribution, the remaining outliers were removed using quantiles, an outlier removal method. After sorting the data by value, the data were divided into quartiles according to the size of each value, and then the outliers were removed using the following Equations (1)–(3).
IQR = Quartile3 − Quartile1 (1)
where IQR = interquartile; Quartile3 = third quartile; Quartile1 = first quartile.
CUO = Quartile3 + 1.5 × IQR (2)
where CUO = criteria for upper outliers.
CLO = Quartile1 − 1.5 × IQR (3)
where CLO = criteria for lower outliers.

Goats lie on their side, stretch their legs, and keep moving when they feel labor pains just before parturition. The triaxial acceleration values of the goats showed an increased number of activities according to their frequent movements caused by labor pains just before parturition, and the number of classifications of labor behavior increased according to their behavior of lying (lateral recumbency) on their side and stretching their legs. Therefore, to classify labor behavior and non-labor behavior using these criteria, the LPI was calculated as shown in Equation (4).
LPI = TA × TLP(4)
where LPI = Labor Pain Index; TA = total number of activities during the time period; TLP = total number of behaviors classified as labor pains by DT during the time period. 

Example: LPI (16,000) =TA (400) × TLP (40)

### 2.7. Selection of the Optimal Time Window

Parturition LPIs were obtained from the highest-value LPIs on the day of parturition for the 17 mother KNBGs. Non-parturition LPIs were calculated by taking the 110 highest-value LPIs from the same goats’ non-parturition behaviors prior to parturition, and the means and standard deviations of these groups were averaged. The central range value (CRV) was calculated using Equations (5)–(7) to classify the parturition and non-parturition LPIs according to the time windows of 8, 10, 12, and 14 min.
MSL = MLPIL − SDLPIL(5)
where MSL = mean minus standard deviation of LPIs during labor pain; MLPIL = mean of LPIs during labor pain; SDLPIL = standard deviation of LPIs during labor pain.
MSNL = MLPINL + SDLPINL(6)
where MSNL = mean plus standard deviation of LPIs during non-labor; MLPINL = mean of LPIs during non-labor; SDLPINL = standard deviation of LPIs during non-labor.
CRV = MSL − MSNL(7)
where CRV = Central range value. 

Among the CRVs obtained using Equation (7), we determined that the time window containing the largest value was the optimal processing time window.

Secondary validation of the difference in LPIs between the day of parturition and the day before parturition was performed to determine whether LPIs could be used to classify parturition versus non-parturition. We calculated the LPIs using data from the 17 KNBGs on the day of parturition and used the highest value as the data for parturition and also calculated the LPIs using data from those same 17 KNBGs one day before parturition and used the highest value as the data for non-parturition. The data were tested for normality using the Kolmogorov–Smirnov test and the difference between the mean values of the LPIs during parturition and non-parturition was verified using an independent T-test. Both tests were performed in SPSS 27 (Released 2020 Version 27.0, Armonk, NY, USA, IBM Corp.).

### 2.8. Calculation of the LPI Threshold for Classifying Parturition

According to the three-sigma method, a value greater than the mean value of a group plus three times the standard deviation can be considered an outlier. Values more significant than the LPI mean value of the non-parturition group plus three times the LPI standard deviation were considered outliers and could be considered included in the parturition group. The threshold value of the LPI was calculated using Equation (8).
TCKG = MLPINL + 3 × SDLPINL (8)
where TCKG = threshold for classifying the parturition of goats. 

Predicting the parturition time requires the difference between each individual’s parturition detection time, the first birth time, and the average of these differences. Therefore, the difference values and the average of these differences were calculated. The predicted parturition time was calculated using Equation (9).
PPT = DPT + MDV (9)
where PPT = predicted parturition time; DPT = detected parturition time; MDV = mean of the difference in values between the detected parturition time and the first birth time.

## 3. Results

### 3.1. Selection of the Main Variables

The gain ratio of the X-axis acceleration value was 0.3233, and the gain ratio of the Z-axis acceleration value was 0.3248, which were both relatively high. Therefore, the X- and Z-axis acceleration values were selected as the main variables, as shown in Figure 3.

### 3.2. Development of the Labor Behavior Classification Model

According to the scatterplot with the main variables, the X- and Z-axis acceleration values, there was an overlap between the labor behavior dataset and the non-labor behavior dataset. Therefore, we decided to use a non-linear classification model because we thought that a linear model could not adequately separate the datasets.

The training results with the non-linear models, SVM and DT, are shown in Table 1 and Table 2. The precision and recall of SVM were 0.99 and 0.93 for the non-labor behavior data and 0.93 and 0.99 for the labor behavior data, respectively, and the precision and recall of DT were 0.96 and 0.97 for the non-labor behavior data and 0.97 and 0.96 for the labor behavior data, respectively. Among them, the accuracy of 0.97 (97.0%) for DT was higher than the accuracy of 0.93 (93.0%) for SVM. As a result, DT was selected as the ML model for labor behavior classification.

### 3.3. Selection of the Optimal Processing Time Window

The results of the CRV calculations are shown in Table 3. The greatest difference was 1690.49 for the 10 min processing time window. In addition, the results of the normality test and the difference between the mean values of the LPIs during labor and non-labor behaviors showed that normality was satisfied (*p* > 0.05), and there was a significant difference between the mean values (*p* < 0.001). Therefore, we concluded that a 10 min processing time window could best classify labor and non-labor behaviors, and we decided to use this time window as the optimal time interval.

### 3.4. Determination of the Threshold for Parturition Classification

The threshold value for parturition was calculated by adding the mean (4807.18) to three times the standard deviation (3144.58) of the 110 highest LPI values of the 10 min processing time window for the 24 h preceding parturition for the 17 KNBGs evaluated, and the result was 14,240.92. Therefore, the threshold value for parturition classification was determined to be 14,240.92.

### 3.5. Calculation of the Parturition Detection Rate

Examples of the results of detecting versus not detecting parturition using the parturition classification threshold are shown in Figure 4 and Figure 5. In Figure 4, all LPIs for non-parturition were below the parturition classification threshold; in contrast, some LPIs for parturition were above the parturition classification threshold and thus could be classified as parturition. However, in Figure 5, all LPIs for non-parturition and parturition were below the parturition classification threshold and could not be classified as parturition.

Out of the 17 KNBGs that underwent parturition, 14 were detected successfully, representing a parturition detection rate of 82.4%; however, 3 were not detected, representing a failure rate of 17.6%. This 82.4% parturition detection rate was very high. In the case of non-parturition, 17 out of 17 goats were successfully detected.

### 3.6. Determination of the Parturition Prediction Time

We calculated the time difference between the time of first birth and the time of the detection of parturition for each individual. We determined the mean and standard deviation of these values; the mean was 90.6 min, and the standard deviation was 99.12 min. Therefore, when detecting parturition, we decided to alert that the first birth would be delivered in an average of 90.6 min based on the time of the detection of parturition. As shown as Figure 6.

## 4. Discussion

The behavioral changes in mother goats in the days before and on the day of labor showed an increase in standing behavior and a decrease in feeding and lying down behavior on the day of giving birth [24]. However, a total of 87% of mother goats exhibit labor pain behaviors such as lying down with their legs stretched out and moving them just before parturition [25]. These results suggest that as labor approaches, the goat becomes less restless and generally exhibits fewer lying behaviors but still exhibits lying behaviors during labor.

The gain ratio of the X- and Z-axis acceleration values measured with the triaxial acceleration sensor was high: the Z-axis acceleration value increased when the goats were lying (lateral recumbency) with their legs stretched out, and the X-axis acceleration value increased with frequent leg movements. These results suggest that the X- and Z-axis acceleration values can be used to measure labor behavior.

In this study, when DT trained a dataset that combined datasets of labor pain and non-labor behaviors, including lying (lateral recumbency) behavior unrelated to labor pain, which were created with X- and Z-axis acceleration values, a classification accuracy of 97% was achieved. Maurmann et al. [20] measured lying behavior in goats not exhibiting labor behavior with a triaxial accelerometer attached to their hind legs and achieved 99.62–99.93% accuracy with video data. These results suggest that triaxial accelerometers attached to the hind limbs of goats can accurately measure lying (lateral recumbency) behavior and distinguish between normal lying behavior (non-labor behavior) and labor behavior.

Goats are also affected by circadian rhythms. Most goats often lie down and get up between 12 p.m. and 2 p.m. and lie down and sleep for a long time between 2 a.m. and 4 a.m. [26]. Therefore, if there is a significant amount of time spent lying during hours usually associated with low levels of lying behavior when measured using either our method or the method of Maurmann et al. [20], we can conclude that certain events, such as illness and parturition, may be occurring. We can also conclude that lying behavior is a marker of rest, thus providing a more accurate indication of welfare. Lying behavior has already been used to assess welfare in cattle [27], sheep [28,29], and goats [30,31]. In particular, lying behavior has been used in dairy goat species as one of the 25 welfare indicators, with abnormal lying posture indicating that the animal may be in pain [32].

The LPI can be measured with a 10 min time window to detect labor behavior and to predict the time of the first parturition. Although the LPIs measured in this study resulted in a high parturition detection rate of 82.4% in KNBGs, the parturition detection rate was calculated from a small number of 17 animals. Therefore, we plan to re-evaluate the parturition detection rate through further studies on KNBG farms.

In this study, of the three (17.6%) goats in which parturition could not be detected, one (5.9%) did not exhibit any lying behavior at all, and the remaining two (11.8%) exhibited brief lying behaviors of less than 1–2 min each. Based on Lickliter [25], 87% of mother goats lie down during labor pains. The slight differences in the proportions of lying behavior during labor pains may be due to differences in breed and management methods.

The parturition prediction time was 90.6 ± 99.12 min before the birth time of the first kid. Thus, the alarm can be noticing the manager to prepare the parturition. Our study did not consider the number of kids delivered as an essential factor. Therefore, we did not detect a correlation between the number of kids delivered and the intensity of labor pain. This potential relationship needs to be investigated in the future.

Future research is also needed to analyze the data from the three goats in which parturition could not be detected to improve the detection rate using variables other than LPI-related variables. In addition, integrated monitoring systems that utilize various sensors and data other than acceleration sensors should be evaluated for their potential utility in this application.

## 5. Conclusions

In this study, we developed a new index LPI based on the labor behavior and activity frequency of KNBGs and a system that classifies labor pain when the LPI is above a threshold and non-labor when the LPI is below. This is a proof-of-concept study, and we will develop an improved version in future studies. The LPI for predicting labor in KNBGs can be applied to labor detection systems for other small ruminants. However, the accuracy of the other species may vary depending on the frequency of the targeted labor behavior.

## Figures and Tables

**Figure 1 animals-14-00634-f001:**
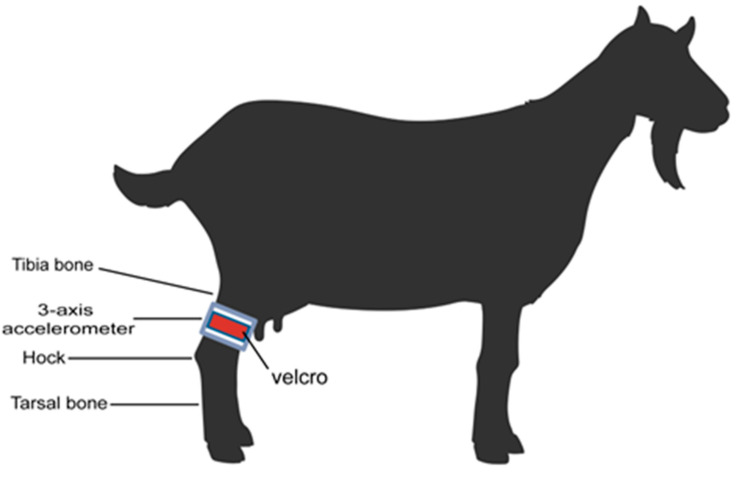
Selecting a sensor attachment location.

**Figure 2 animals-14-00634-f002:**
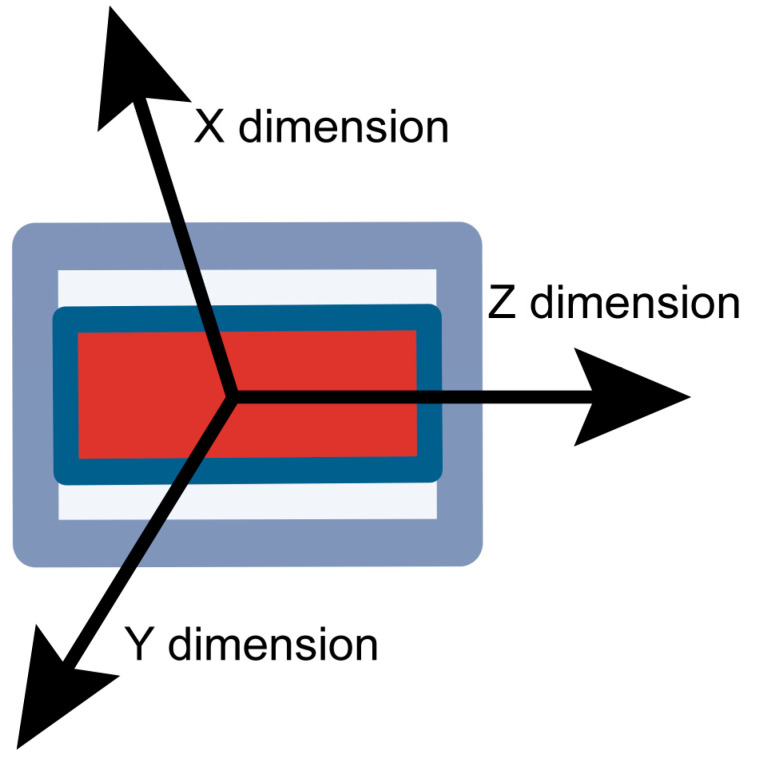
Dimensions of triaxial acceleration sensor.

**Figure 3 animals-14-00634-f003:**
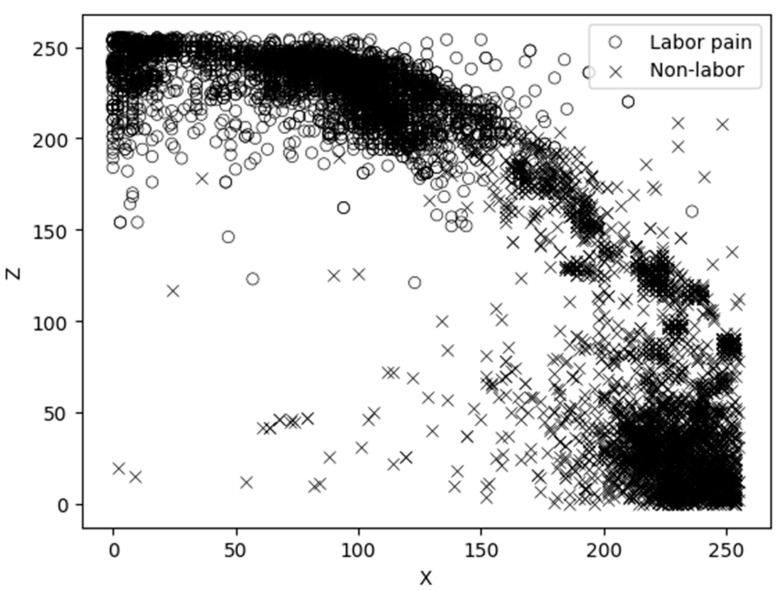
Scatter plot of dataset 1 for labor behavior and dataset 2 for non-labor behavior.

**Figure 4 animals-14-00634-f004:**
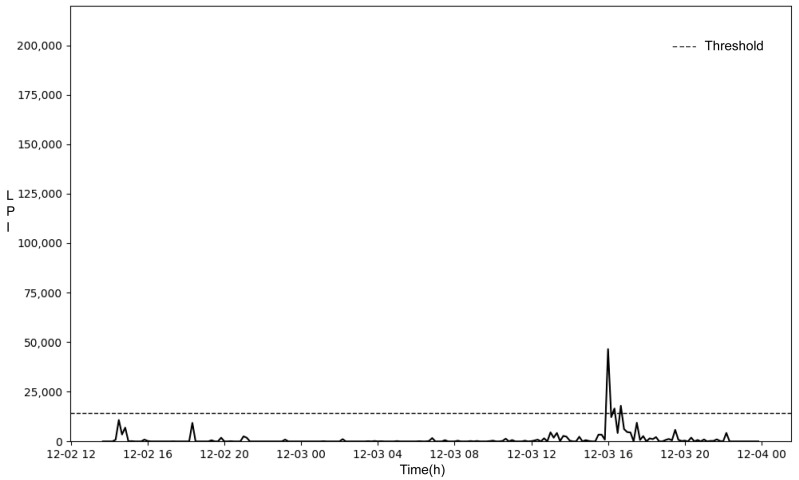
Parturition detection data.

**Figure 5 animals-14-00634-f005:**
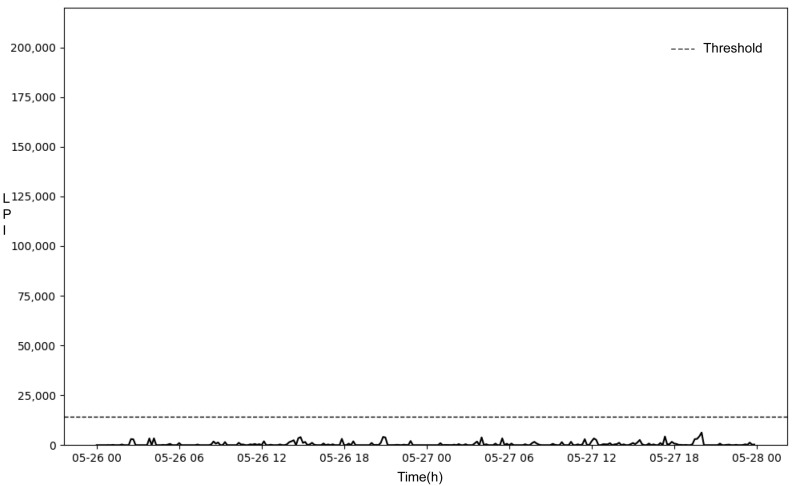
Not classified as parturition.

**Figure 6 animals-14-00634-f006:**
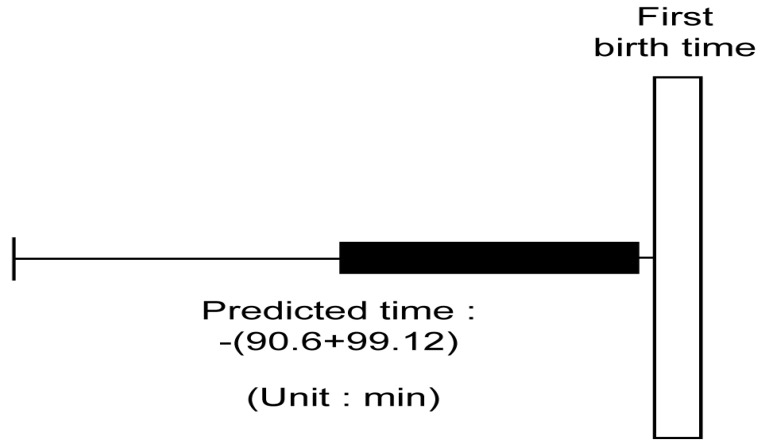
Predicted time of first birth.

**Table 1 animals-14-00634-t001:** Accuracy of Support Vector Machine.

	Precision	Recall	F-1 Score	Support
Non-labor (0)	0.99	0.93	0.96	782
Labor (1)	0.93	0.99	0.96	794
Accuracy			0.96	1576
Macro average	0.96	0.96	0.96	1576
Weightedaverage	0.96	0.96	0.96	1576

Precision: true positive ÷ (true positive + false positive). Recall: true positive ÷ (true positive + false negative). F-1 score: 2 × (precision × recall) ÷ (precision + recall). Support: number of the quantities. Macro average: sum(F1 scores) ÷ number of classes. Weighted average: ∑(weights × quantities) ÷ ∑Weights.

**Table 2 animals-14-00634-t002:** Accuracy of Decision Tree.

	Precision	Recall	F-1 Score	Support
Non-labor (0)	0.96	0.97	0.96	782
Labor (1)	0.97	0.96	0.96	794
Accuracy			0.96	1576
Macro average	0.96	0.96	0.96	1576
Weightedaverage	0.96	0.96	0.96	1576

Precision: true positive ÷ (true positive + false positive). Recall: true positive ÷ (true positive + false negative). F-1 score: 2 × (precision × recall) ÷ (precision + recall). Support: number of the quantities. Macro average: sum(F1 scores) ÷ number of classes. Weighted average: ∑(weights × quantities) ÷ ∑weights.

**Table 3 animals-14-00634-t003:** Means and standard deviations of LPIs during labor and non-labor periods for KNBGs.

Time Window (min)	Labor			Non-Labor			A–B	Significance
	Mean	Standard Deviation	Mean − Standard Deviation (A)	Mean	Standard Deviation	Mean + Standard Deviation (B)		
8	69,359.59	66,253.97	3105.62	3550.65	1781.54	5332.18	−2226.57	*p* < 0.001
10	77,780.06	68,137.81	9642.25	4807.18	3144.58	7951.75	1690.49	*p* < 0.001
12	102,674.63	99,319.32	3355.31	6248.59	3222.90	9471.49	−6116.18	*p* < 0.001
14	107,822.07	102,600.46	5221.60	7547.88	4149.09	11.696.97	−6475.37	*p* < 0.001

## Data Availability

The data were not deposited in an official repository. The data supporting the findings are available upon request from the corresponding author (H.C.).

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
