# Peer review of "Development of a Parturition Detection System for Korean Native Black Goats"

_animals, 2024, doi:10.3390/ani14040634_

Round 1
Reviewer 1 Report
Comments and Suggestions for Authors
General comment:
The study by Kim et al. evaluated an accelerometer-based parturition detection system in goats, showcasing significance amid the advancements in precision livestock tools. The developed system demonstrates acceptable accuracy in detecting parturition time in goats and holds practical implications for applications in goat production. The manuscript was well-written and easy to follow. Additional feedback is provided below to enhance the manuscript's quality.
Abstract
The abstract exceeds the recommended length for the journal. Please, consider reducing it to approximately 250 words.
Introduction
Lines 59-63: The information appears redundant and obstructs the flow of the introduction section. Please consider removing these sentences. Instead, introduce a sentence highlighting the use of accelerometers to detect behavioral changes, and reinforce this statement with additional examples, as you have already provided.
Material and Methods
Line 91: Since there is no description of the experimental design, this subheading should be 'Animals and Housing.
Line 127: If the word 'Velcro' refers to a general double-sided hook and loop fastener tape, there is no need to capitalize it.
Line 135-141. "Lines 135-141: Consider adding a picture of a goat with sensors attached for visual illustration. Additionally, provide the total weight of the wearable sensor for completeness.
Lines 150-153: Consider relocating this paragraph to the end of Section 2.3 for better organization and flow.
The data collection and analysis processes were well-documented, contributing to the overall rigor of the study.
Results and Discussion
Tables 1 and 2: Please use the full words in the table titles for SVM (Support Vector Machine) and DT (Decision Tree).
Line 319: Correct the figure numbers referenced in the text."
Figures 2 and 3: Please ensure that the details of the figures, particularly the threshold line (better to have this line dotted), are appropriately added to the captions for better clarity.
Lines 341-345: Since this study did not assess hormonal concentrations during parturition, it would be better not to include such information to begin the discussion. Instead, focus on behavioral changes before parturition for a more coherent narrative.
The discussion was well-supported with existing literature and effectively identified some of the limitations of the study like sample size. Moreover, it offered insightful suggestions for future research prospects aimed at enhancing the system.
Conclusion
Lines 407-409: The sentence is challenging to understand. Please summarize your key findings in a clear and impactful manner for improved comprehension.
Author Response
Dear journal editor and reviewers,
We are very grateful to the editor for your appropriate and constructive suggestions and for proposed correction to improve the paper quality. We also thank for the effort and time put into the review of the manuscript. The editor and reviewers have brought up some good points and we appreciate the opportunity to clarify our work objective. We tried our best to respond the concerns. The major corrections are listed below point by point. The revised texts are highlighted in red color in the revised manuscript.
- Reviewer 1
The study by Kim et al. evaluated an accelerometer-based parturition detection system in goats, showcasing significance amid the advancements in precision livestock tools. The developed system demonstrates acceptable accuracy in detecting parturition time in goats and holds practical implications for applications in goat production. The manuscript was well-written and easy to follow. Additional feedback is provided below to enhance the manuscript's quality.
Abstract
The abstract exceeds the recommended length for the journal. Please, consider reducing it to approximately 250 words.
We changed the abstract as you suggested, shortening it to 225 words.
Introduction
Lines 59-63: The information appears redundant and obstructs the flow of the introduction section. Please consider removing these sentences. Instead, introduce a sentence highlighting the use of accelerometers to detect behavioral changes, and reinforce this statement with additional examples, as you have already provided.
We've deleted the part you mentioned to make the sentence flow more directly.
Adding new paragraph in Line 51~55
Material and Methods
Line 91: Since there is no description of the experimental design, this subheading should be 'Animals and Housing.
The word Animals and experimental design’ has been changed into ’Animals and Housing’, as you suggested.
Line 127: If the word 'Velcro' refers to a general double-sided hook and loop fastener tape, there is no need to capitalize it.
We changed the V in line 127 to v as you suggested.
The line has changed and is now on line 124.
Line 135-141. "Lines 135-141: Consider adding a picture of a goat with sensors attached for visual illustration. Additionally, provide the total weight of the wearable sensor for completeness.
Added an goat with sensors attached for visual illustration on line 149.
The sensor weighs 125 grams, and you have indicated on line 118 that the sensor is 25 grams and the plastic housing is 100 grams.
Lines 150-153: Consider relocating this paragraph to the end of Section 2.3 for better organization and flow.
Paragraph moved from line 114 to 118.
The data collection and analysis processes were well-documented, contributing to the overall rigor of the study.
Results and Discussion
Tables 1 and 2: Please use the full words in the table titles for SVM (Support Vector Machine) and DT (Decision Tree).
We've explained it in detail as you said.
Line 319: Correct the figure numbers referenced in the text."
In line 332, 335 we fix the mismatch between the picture and the numbering as you mentioned.
Figure 8 -> 4, Figure 9 -> 5
Figures 2 and 3: Please ensure that the details of the figures, particularly the threshold line (better to have this line dotted), are appropriately added to the captions for better clarity.
In line 342 and 344, we’ve fixed it with a dotted line and added a caption.
Lines 341-345: Since this study did not assess hormonal concentrations during parturition, it would be better not to include such information to begin the discussion. Instead, focus on behavioral changes before parturition for a more coherent narrative.
We've removed the parts and the add the new paragraph in line 364~367, 369~371
The discussion was well-supported with existing literature and effectively identified some of the limitations of the study like sample size. Moreover, it offered insightful suggestions for future research prospects aimed at enhancing the system.
Conclusion
Lines 407-409: The sentence is challenging to understand. Please summarize your key findings in a clear and impactful manner for improved comprehension.
We realized that the conclusion wasn't clear enough, so we changed the semantics to make it clearer.
Reviewer 2 Report
Comments and Suggestions for Authors
animals-2851053
Title: Development of a parturition detection system for Korean native black goats
Authors: Heungsu Kim, Hyunse Kim, Woohyun Kim, Wongi Min, Geonwoo Kim , Honghee Chang
First of all, I would like to thank the authors for putting in the time to present a grammatically well written manuscript. There are a few places I will point out specifically where the translation into English could be improved, but overall well-written.
Overall, I think the presentation in the manuscript needs more detail to communicate to the reader what was actually done, and more importantly why. For instance, the manuscript describes the use of only 17 KNBG goat parturition events. Then, there is indication that the data were split into a learning set (80%) and a testing set (20%). But this was not separately by goat, thus the population that prediction is assessed for is only these 17 parturition events and the large numbers of observations are not independent of each other. Predictability can only truly be considered if the resulting methods is applied to a different set of goats, preferably in a different herd. Thus, the determined accuracy is not surprising, but also not convincing. Also, I do not believe that the results based on only 17 goats support the very broad conclusions reached until further testing is completed. The work should be portrayed more as a proof of concept than actual development of the equations for predicting labor pains and time of first birth.
Also, it appears that labor behavior and labor pain are used interchangeably throughout the manuscript, but it was not mentioned (unless I missed it) how the levels of pain were determined.
Line Comment
11 Delete “Therefore”. The conclusion does not depend on the first sentence.
13-14 …”applying this system to other Korean native black goats or other breeds of goats will require…”
17 “Because” not “As”
19 Suggest “Managing parturition is a way…” The word “nursing has connotations of getting the kid to nurse on the doe.
20 “managers” instead of “tenders”
26 Here and throughout the manuscript, please use “3,938” style instead of “3 938”
27 How do you know that the does felt labor pains? Was pain measured objectively, or could this simply have been a change in behavior?
30-31 How was the data classified into the processing time windows?
31 Was the activity count data within the specific processing time windows? What is an “activity count value”?
34 This is a problem with many precision technologies. If the error rate is as high as 3 of 17 parturitions, managers will grow tired of false alarms and ignore the signals. This error rate seems especially unsatisfying because the results were applied only to the parturitions on which the thresholds were determined.
37 Need to explain why the average detection time was 90.6 minutes before parturition, but the promise is that it can be detected 25.6 minutes before. Need to reconcile this discrepancy.
39 Keywords…this has little to do with goat kid and more with does. “management” would be better term than “nursing”. It seems this is more about labor behaviour than labor pain, since that was not really measured.
55 “lying down” is a confusing term because it is used interchangeably with lying and can include both sternal and lateral recumbency. This should be fixed throughout.
78 Who does “their” refer to?
86 delete “pain”
94 17 seems like a very small number to allow the sweeping conclusions you reach.
134-141 A diagram to describe the coordinates would be best, especially to describe the potential motions of a doe lying laterally on her side. It is difficult to picture which dimension would be most affected. It seems correct, but difficult to picture when the posture changes.
177 How was this verification that the does were experiencing labor pain performed?
178 Why did the parturition and prior day values use the same number of data points and not the same time duration? Was paired data required in the analyses? If data points represented 10 second intervals when no motion, one would expect the elapsed time for collecting the 3,938 data points to occur over very different time periods.
186 How were the data points assigned to the training and test data? That is critical, since the datapoints are not independent observations.
205 “…were calculated for each of these periods.”
207 Delete either “of” or “from”
215 It is unclear what you mean by “decimal” in each of the equations. If it just means the numbers are not rounded to whole units, it is not needed.
234 A brief example of the calculations would be very helpful.
236 Delete the first 17 in the sentence. Sounds like 17 for each doe.
265 True, but mean plus or minus 3 SDs is the standard from Central Distribution Theory, but is a very limiting standard to define outliers of biological processes. Because the 3 SDs are expected to cover 98% of datapoints, that leaves 1% of datapoints (one tail) that would be expected values now declared as outliers when they are expected outcomes.
276 How is the detected parturition time different from the first birth time? Seems like MDV would always be zero.
281 Need to define z and x in the figure cation. Also, use labor behavior and not labor pain. It might also be wise to indicate the level of change in y compared to z and x.
299 Write out SVM for the Table heading. Also need to define Recall, F-1 Score, Support, Macro Average, and weighted average in the heading or Table footnotes. Same for DT and other items in Table 2.
310 Reformat Table 3 with smaller font size. What test is used for the significance test for which the probabilities are reported. Because the means are very disparate and one is plus a standard deviation and the other minus, is a test of significance even needed?
216 Was this number consistent if different datapoints were used in the learning and testing data points?
319 I could not find Figures 8 and 9, so unclear what this is referring to? Figures 1 and 2, perhaps?
326 If the manager is alerted and 17% of the time the parturition has not started, they will soon stop responding to the alerts.
331 According to the description, this appears to “Not classified as parturition”. So the Figure 3 caption should be changed.
337 On average is not very convincing. If the average is 90.6 minutes and the SD is 99.12 minutes, then the expected delivery (95% Confidence Interval) would be somewhere between -107 and 289 minutes which would not be very useful as an alert.
338 This appears to be the very first mention of primiparous and multiparous goats. This should be reported in the Materials and Methods. How many of each were there? Were the number of data points the same for each group between parturition and the day before? Judging by the SD shown in the whisker plot, the number of primiparous goats must have much larger than multiparous. With only 17 goats total, there cannot be much confidence in this resulting difference.
348 Adrenaline is well known to interfere with oxytocin receptors in myoepithelial cells related to milk ejection. Does the same happen in the birth canal. Could differences in adrenaline then also affect the results?
354 to 357 Where is this shown in your results?
363-375 How were circadian rhythms manifested or considered in this research?
Comments on the Quality of English Language
See in Comments to authors.
Author Response
Dear journal editor and reviewers,
We are very grateful to the editor for your appropriate and constructive suggestions and for proposed correction to improve the paper quality. We also thank for the effort and time put into the review of the manuscript. The editor and reviewers have brought up some good points and we appreciate the opportunity to clarify our work objective. We tried our best to respond the concerns. The major corrections are listed below point by point. The revised texts are highlighted in red color in the revised manuscript.
Reviewer 2
First of all, I would like to thank the authors for putting in the time to present a grammatically well written manuscript. There are a few places I will point out specifically where the translation into English could be improved, but overall well-written.
Overall, I think the presentation in the manuscript needs more detail to communicate to the reader what was actually done, and more importantly why. For instance, the manuscript describes the use of only 17 KNBG goat parturition events. Then, there is indication that the data were split into a learning set (80%) and a testing set (20%). But this was not separately by goat, thus the population that prediction is assessed for is only these 17 parturition events and the large numbers of observations are not independent of each other. Predictability can only truly be considered if the resulting methods is applied to a different set of goats, preferably in a different herd. Thus, the determined accuracy is not surprising, but also not convincing. Also, I do not believe that the results based on only 17 goats support the very broad conclusions reached until further testing is completed. The work should be portrayed more as a proof of concept than actual development of the equations for predicting labor pains and time of first birth.
Also, it appears that labor behavior and labor pain are used interchangeably throughout the manuscript, but it was not mentioned (unless I missed it) how the levels of pain were determined.
First, to answer your question about the 80:20 split of the 17 birthing events, I used an 80:20 split of the 3938 data from the 17 birthing events with pain and the 3938 data from the 3938 without pain. I did not use the goats' data individually because there were certain behaviors that goats had in common before parturition. In order to observe and generalize from these behaviors, I needed to lump them together as an overall value to create a data set that represents the population's behavior as a whole.
While it might seem challenging to average a population from 17 data points, the number of purebred Korean black goats is minimal, so we collected data from a single farm for two years from November 19 to May 21, and the number of data points collected was 17, excluding erroneous sensor data.
National organizations utilize most purebred Korean black goats as genetic resources, so convincing data is needed to apply to other goat herds through further experiments. We published this paper to secure data from other Korean black goats. In addition, the data from the experiment with 17 Korean black goats cannot support broad conclusions, but it can be said to be a review of observations obtained through my observations of goats over the years.
Also, you mentioned that it was more of a proof of concept than an actual development of an equation, and you are right. In this research, we tried to define labor behavior through machine learning, then identify the difference between labor and non-labor through the labor pain index for the actual behavior that occurs during labor, and then show that the actual development will be verified in the field after securing several groups of goats in a follow-up study.
In addition, the indication of labor pain was defined as a goat lying down completely and howling or stretching its legs. However, we did not evaluate the intensity of labor pain. However, we thought that the intensity of labor pain was more potent if the goat felt labor pain for a more extended period than other goats at the time of delivery, so we expressed the pain intensity in the paper, but we will modify it if necessary.
Line Comment
11 Delete “Therefore”. The conclusion does not depend on the first sentence.
We've crossed out what you said on line 11.
13-14 …”applying this system to other Korean native black goats or other breeds of goats will require…”
We've modified line 14 to fit the context as you suggested.
17 “Because” not “As”
Removed from the abstract.
19 Suggest “Managing parturition is a way…” The word “nursing has connotations of getting the kid to nurse on the doe.
Removed from the abstract.
20 “managers” instead of “tenders”
We've made the changes you mentioned on line 18.
26 Here and throughout the manuscript, please use “3,938” style instead of “3 938”
Removed from the abstract.
27 How do you know that the does felt labor pains? Was pain measured objectively, or could this simply have been a change in behavior?
“ The externally visible signs of the first stage of labor in the cow, buffalo, ewe and goat include symptoms of mild abdominal pain, frequent getting up and lying down which are marked in the primiparous animals.”
“Almost all animals lie down as soon as straining commences. Occasionally the foal or calf may be born with the dam standing. The mare and the sow usually lie out in lateral position with legs extended, whereas the cow, bitch and ewe are more likely to lie on their sternum.”
Purohit, G. (2010). Parturition in domestic animals: A review.
doi: 10.9754/journal.wmc.2010.00748
“The behavioral characteristics of Norduz primiparous does in pre-parturition are shown in Table 1. Twelve single-birth does (67%) gave birth while recumbent, the remaining six (33%) gave birth standing. There is significant differences regarding the shape of birth “
Yilmaz, A., Karaca, S., Kor, A., & BINGOL, M. (2012). Determination of pre-parturition and post-parturition behaviors of Norduz goats. Kafkas Üniversitesi Veteriner Fakültesi Dergisi, 18(2).
30-31 How was the data classified into the processing time windows?
This is because the large difference between labor and non-labor among the time windows demonstrates the difference between the labor and non-labor groups and shows that the two groups can be clearly separated. The data from labor and non-labor were subtracted by multiplying the number of contractions and behaviors present in the time window, and the most significant difference was considered the optimal processing time.
31 Was the activity count data within the specific processing time windows? What is an “activity count value”?
The "activity count value" is about the value of "activity" present in the 3-axis acceleration data. when the motion is detected, the sensor shows the "activity" state. Then, the number of such "activities" accumulated and present within the time window is the "activity count value."
34 This is a problem with many precision technologies. If the error rate is as high as 3 of 17 parturitions, managers will grow tired of false alarms and ignore the signals. This error rate seems especially unsatisfying because the results were applied only to the parturitions on which the thresholds were determined.
We'll do further experiments to see how we can lower the error rate in the field and find a threshold for detection with fewer errors.
37 Need to explain why the average detection time was 90.6 minutes before parturition, but the promise is that it can be detected 25.6 minutes before. Need to reconcile this discrepancy.
That was a mistake on my part. I corrected it to 90.6 on line 29.
39 Keywords…this has little to do with goat kid and more with does. “management” would be better term than “nursing”. It seems this is more about labor behaviour than labor pain, since that was not really measured.
We have changed the keyword you mentioned.
55 “lying down” is a confusing term because it is used interchangeably with lying and can include both sternal and lateral recumbency. This should be fixed throughout.
We have notated after lying where conceptually it should be described as lateral recumbency. In line 140, 171, 172, 206, 229, 369, 373 and 378.
78 Who does “their” refer to?
The "their" you're referring to is a mother black goat in labor.
86 delete “pain”
We've fixed it as you suggested.
94 17 seems like a very small number to allow the sweeping conclusions you reach.
That parturition detection system using a raider sensor, and them uses 18 sows.
Manteuffel, C. (2019). Parturition detection in sows as test case for measuring activity behaviour in farm animals by means of radar sensors. Biosystems Engineering, 184, 200-206. https://doi.org/10.1016/j.biosystemseng.2019.06.018
That parturition detection system using an accelerometer, and them uses 19 sows.
Cornou, C., & Lundbye-Christensen, S. (2012). Modeling of sows diurnal activity pattern and detection of parturition using acceleration measurements. Computers and electronics in agriculture, 80, 97-104. https://doi.org/10.1016/j.compag.2011.11.001
That parturition detection system only an accelerometer and uses 17 cows.
Chang, A. Z., Fogarty, E. S., Swain, D. L., García-Guerra, A., & Trotter, M. G. (2022). Accelerometer derived rumination monitoring detects changes in behaviour around parturition. Applied Animal Behaviour Science, 247, 105566. https://doi.org/10.1016/j.applanim.2022.105566
134-141 A diagram to describe the coordinates would be best, especially to describe the potential motions of a doe lying laterally on her side. It is difficult to picture which dimension would be most affected. It seems correct, but difficult to picture when the posture changes.
Figure 1 shows the attachment location of the sensor on line 145, and figure 2 shows the dimensions of the 3-axis accelerometer on line 146.
177 How was this verification that the does were experiencing labor pain performed?
“ The externally visible signs of the first stage of labor in the cow, buffalo, ewe and goat include symptoms of mild abdominal pain, frequent getting up and lying down which are marked in the primiparous animals.”
“Almost all animals lie down as soon as straining commences. Occasionally the foal or calf may be born with the dam standing. The mare and the sow usually lie out in lateral position with legs extended, whereas the cow, bitch and ewe are more likely to lie on their sternum.”
Purohit, G. (2010). Parturition in domestic animals: A review.
doi: 10.9754/journal.wmc.2010.00748
“The behavioral characteristics of Norduz primiparous does in pre-parturition are shown in Table 1. Twelve single-birth does (67%) gave birth while recumbent, the remaining six (33%) gave birth standing. There is significant differences regarding the shape of birth “
Yilmaz, A., Karaca, S., Kor, A., & BINGOL, M. (2012). Determination of pre-parturition and post-parturition behaviors of Norduz goats. Kafkas Üniversitesi Veteriner Fakültesi Dergisi, 18(2).
178 Why did the parturition and prior day values use the same number of data points and not the same time duration? Was paired data required in the analyses? If data points represented 10 second intervals when no motion, one would expect the elapsed time for collecting the 3,938 data points to occur over very different time periods.
We used the same number of data points for labor phase and the day before labor phase.
because, using the same number of data points, rather than time, would be more representative of non-labor behavior when evaluated with the overall number of data points.
We used the same number of data points to include more indicators of rest, as non-labor animals are more likely to rest rather than continue to move vigorously, compared to labor animals, which are more likely to continue to move while feeling contractions.
186 How were the data points assigned to the training and test data? That is critical, since the datapoints are not independent observations.
The data points were randomly assigned: 3938 labor and non-labor data were split 80% to 20%.
205 “…were calculated for each of these periods.”
We've added a sentence to line 204 as you suggested, so it doesn't make sense out of context.
207 Delete either “of” or “from”
For the part you mentioned on line 206, we deleted from between Of and from.
215 It is unclear what you mean by “decimal” in each of the equations. If it just means the numbers are not rounded to whole units, it is not needed.
Delete “decimal” in line 217, 218, 219, 221, 223, 232, 233, 235, 244, 245,246, 248, 249, and 252.
234 A brief example of the calculations would be very helpful.
We tried writing the example expression on line 230 as you suggested.
Example : LPI(16000)=TA(400) × TLP(40)
236 Delete the first 17 in the sentence. Sounds like 17 for each doe.
We've fixed it as you suggested.
265 True, but mean plus or minus 3 SDs is the standard from Central Distribution Theory, but is a very limiting standard to define outliers of biological processes. Because the 3 SDs are expected to cover 98% of datapoints, that leaves 1% of datapoints (one tail) that would be expected values now declared as outliers when they are expected outcomes.
Yes, that is correct, but we were concerned that not removing the 1% extreme values would skew the study results, so we used the 3SD method of outlier removal.
276 How is the detected parturition time different from the first birth time? Seems like MDV would always be zero.
The detected time of labor is the part of the entire labor process where the mother goat feels the pain of lying down and feeling the pain, and the time of the first kid is the time of expulsion after feeling the pain of labor.
281 Need to define z and x in the figure cation. Also, use labor behavior and not labor pain. It might also be wise to indicate the level of change in y compared to z and x.
After analyzing the XYZ values through WEKA analysis, it was determined that X and Z values were suitable for observing the goat's labor behavior, so the Y value was excluded. Also, the accuracy of the machine learning model to classify labor decreased when applying the Y value, so it was excluded.
299 Write out SVM for the Table heading. Also need to define Recall, F-1 Score, Support, Macro Average, and weighted average in the heading or Table footnotes. Same for DT and other items in Table 2.
In Table 1, we added the definitions below for Recall, F-1 Score, Support, Macro Average, and weighted average mentioned in lines 296-302. In Table 2, we have added definitions for Recall, F-1 Score, Support, Macro Average, weighted average, and weighted average, as mentioned in lines 303-308.
310 Reformat Table 3 with smaller font size. What test is used for the significance test for which the probabilities are reported. Because the means are very disparate and one is plus a standard deviation and the other minus, is a test of significance even needed?
As you suggested, we changed the font size from 10 to 8, used an independent t-test for significance testing to report probabilities, and while there is clearly a difference between large and small, we have included the p-value to make the difference clear.
216 Was this number consistent if different datapoints were used in the learning and testing data points?
We set the randomization state to 42 to ensure that the data frames are equally mixed, and we set the depth value of the decision tree to 5 to avoid overfitting.
319 I could not find Figures 8 and 9, so unclear what this is referring to? Figures 1 and 2, perhaps?
Change the figure number in line 332 and 335.
Figure 8 -> 4, Figure 9 -> 5.
326 If the manager is alerted and 17% of the time the parturition has not started, they will soon stop responding to the alerts.
It is not a false alarm 17% of the time; it is a missed detection 17% of the time. The alert is not 100% accurate, but it can be used to help administrators when it is difficult to manage.
331 According to the description, this appears to “Not classified as parturition”. So the Figure 3 caption should be changed.
We've made the changes you mentioned.
In line 345.
337 On average is not very convincing. If the average is 90.6 minutes and the SD is 99.12 minutes, then the expected delivery (95% Confidence Interval) would be somewhere between -107 and 289 minutes which would not be very useful as an alert.
Because the SD is larger than the mean, there may be instances where the notification time is too early or too late. However, in future study this problem can be solved by increasing the population to reduce the impact of extreme values.
338 This appears to be the very first mention of primiparous and multiparous goats. This should be reported in the Materials and Methods. How many of each were there? Were the number of data points the same for each group between parturition and the day before? Judging by the SD shown in the whisker plot, the number of primiparous goats must have much larger than multiparous. With only 17 goats total, there cannot be much confidence in this resulting difference.
The number of primiparous goats was 5, and 9 multiparous goats, although this is not provided in Materials and Methods as we only used individuals with successful estrus detection. The overall number of data points was different on the day before parturition than it was on the day of parturition. However, we believe that this is due to the effect of anxiety and movement during parturition. For primiparous, five goats were detected to be in labor 5 hours before parturition, and for multiparous, 3 of the nine detected goats were not detected to be in labor. Moreover, when trying to present the results in aggregate, the number of goats is small at 17, so I wonder if it would be better to delete the primiparous and multiparous goats and combine them and present the total data?
348 Adrenaline is well known to interfere with oxytocin receptors in myoepithelial cells related to milk ejection. Does the same happen in the birth canal. Could differences in adrenaline then also affect the results?
https://www.mdpi.com/animals/animals-11-02960/article_deploy/html/images/animals-11-02960-g003.png
Adrenaline affect the oxytocin level as a inhibiting indicator. But in birth canal did not have any information in a google scholar and pubmed, I suspect it will have an impact, but I don't know for sure.
354 to 357 Where is this shown in your results?
The result in table 2, located in section 3.2. line 308.
363-375 How were circadian rhythms manifested or considered in this research?
There is nothing directly about circadian rhythms in this study, but the main argument was that lying down behavior itself is affected by circadian rhythms, like the adrenaline that the reviewer mentioned, so that should be taken into account as well.
Reviewer 3 Report
Comments and Suggestions for Authors
General comments:
It is a meticulous job with objectives that are clearly defined. The design and methodology are correct and the bibliographic review is adequate and up-to-date. It is well designed and well crafted, with small observations. The results show that the first 3 objectives have been achieved, with a birth detection rate of 82.4%, but nothing is commented on the results of the 4th objective (Develop a notification system that can provide early warning for labor), which would be very interesting for the practical application of the research carried out in the work.
The manuscript represents a step forward for the application of precision livestock farming in small ruminants. Some observations are discussed below.
Specific comments:
Lines 319 and 322. Figures 8 and 9 do not exist. Review.
Figures 2, 3 and 4 are not cited in the text. Review.
Author Response
Dear journal editor and reviewers,
We are very grateful to the editor for your appropriate and constructive suggestions and for proposed correction to improve the paper quality. We also thank for the effort and time put into the review of the manuscript. The editor and reviewers have brought up some good points and we appreciate the opportunity to clarify our work objective. We tried our best to respond the concerns. The major corrections are listed below point by point. The revised texts are highlighted in red color in the revised manuscript.
Reviewer 3
It is a meticulous job with objectives that are clearly defined. The design and methodology are correct and the bibliographic review is adequate and up-to-date. It is well designed and well crafted, with small observations. The results show that the first 3 objectives have been achieved, with a birth detection rate of 82.4%, but nothing is commented on the results of the 4th objective (Develop a notification system that can provide early warning for labor), which would be very interesting for the practical application of the research carried out in the work.
We've made the changes you mentioned.
The manuscript represents a step forward for the application of precision livestock farming in small ruminants. Some observations are discussed below.
Specific comments:
Lines 319 and 322. Figures 8 and 9 do not exist. Review.
We've made the changes you mentioned.
Figures 2, 3 and 4 are not cited in the text. Review.
We've made the changes you mentioned.
Round 2
Reviewer 2 Report
Comments and Suggestions for Authors
Thank you for the improvements to the manuscript.
I agree that it would be better to only focus on the overall kiddings and not split out by parity. That really magnifies the limited number of KNBGs available for this work.
Please reiterate in the conclusions that this is a proof of concept study and that models will be tested with additional animals in the future.
Author Response
Dear journal editor and reviewers,
We are very grateful to the editor for your appropriate and constructive suggestions and for proposed correction to improve the paper quality. We also thank for the effort and time put into the review of the manuscript. The editor and reviewers have brought up some good points and we appreciate the opportunity to clarify our work objective. We tried our best to respond the concerns. The major corrections are listed below point by point. The revised texts are highlighted in red color in the revised manuscript.
Thank you for the improvements to the manuscript.
I agree that it would be better to only focus on the overall kiddings and not split out by parity. That really magnifies the limited number of KNBGs available for this work.
We change you mentioned part.
Delete line 353~356 in result.
Delete line 412~415, 416~421 in discussion.
Please reiterate in the conclusions that this is a proof of concept study and that models will be tested with additional animals in the future.
We add this sentence in line 434~435.
This is a proof-of-concept study, and we will develop an improved version in future studies.